# Understanding the Active Mechanisms of Plant (*Sesuvium portulacastrum* L.) against Heavy Metal Toxicity

**DOI:** 10.3390/plants12030676

**Published:** 2023-02-03

**Authors:** Emad A. Alsherif, Mohammad Yaghoubi Khanghahi, Carmine Crecchio, Shereen Magdy Korany, Renato Lustosa Sobrinho, Hamada AbdElgawad

**Affiliations:** 1Biology Department, College of Science and Arts at Khulis, University of Jeddah, Jeddah 21959, Saudi Arabia; 2Botany and Microbiology Department, Faculty of Science, Beni-Suef University, Beni-Suef 62521, Egypt; 3Department of Soil, Plant and Food Sciences, University of Bari Aldo Moro, Via Amendola 165/A, 70126 Bari, Italy; 4Department of Biology, College of Science, Princess Nourah bint Abdulrahman University, Riyadh 11671, Saudi Arabia; 5Department of Agronomy, Federal University of Technology—Paraná (UTFPR), Pato Branco 85503-390, PR, Brazil; 6Integrated Molecular Plant Physiology Research, Department of Biology, University of Antwerp, 2000 Antwerp, Belgium

**Keywords:** amino acid, antioxidant, fatty acid, heavy metal, metabolite, *Sesuvium portulacastrum* L., tocopherol

## Abstract

Through metabolic analysis, the present research seeks to reveal the defense mechanisms activated by a heavy metals-resistant plant, *Sesuvium portulacastrum* L. In this regard, shifting metabolisms in this plant were investigated in different heavy metals-contaminated experimental sites, which were 50, 100, 500, 1000, and 5000 m away from a man-fabricated sewage dumping lake, with a wide range of pollutant concentrations. Heavy metals contaminations in contaminated soil and their impact on mineral composition and microbial population were also investigated. The significant findings to emerge from this research were the modifications of nitrogen and carbon metabolisms in plant tissues to cope with heavy metal toxicity. Increased plant amylase enzymes activity in contaminated soils increased starch degradation to soluble sugars as a mechanism to mitigate stress impact. Furthermore, increased activity of sucrose phosphate synthase in contaminated plants led to more accumulation of sucrose. Moreover, no change in the content of sucrose hydrolyzing enzymes (vacuolar invertase and cytosolic invertase) in the contaminated sites can suggest the translocation of sucrose from shoot to root under stress. Similarly, although this study demonstrated a high level of malate in plants exposed to stress, caution must be applied in suggesting a strong link between organic acids and the activation of defense mechanisms in plants, since other key organic acids were not affected by stress. Therefore, activation of other defense mechanisms, especially antioxidant defense molecules including alpha and beta tocopherols, showed a greater role in protecting plants from heavy metals stress. Moreover, the increment in the content of some amino acids (e.g., glycine, alanine, glutamate, arginine, and ornithine) in plants under metal toxicity can be attributed to a high level of stress tolerance. Moreover, strategies in the excitation of the synthesis of the unsaturated fatty acids (oleic and palmitoleic) were involved in enhancing stress tolerance, which was unexpectedly associated with an increase in the accumulation of palmitic and stearic (saturated fatty acids). Taken together, it can be concluded that these multiple mechanisms were involved in the response to stress which may be cooperative and complementary with each other in inducing resistance to the plants.

## 1. Introduction

Accumulation of heavy metals and potentially toxic elements in soil (e.g., chromium (Cr), mercury (Hg), lead (Pb), cadmium (Cd), arsenic (As), aluminum (Al), thallium (Tl), etc.) is very toxic to plants, even at low concentrations, especially because a majority of them have no role in metabolic reactions in plants [1]. Although some of these elements (e.g., zinc (Zn), copper (Cu), molybdenum (Mo), manganese (Mn), nickel (Ni), etc.) are known as micronutrients and are involved in several crucial biochemical and physiological reactions in plants, they can be toxic to the plant in concentrations exceeding the threshold [2,3]. One of the main causes of heavy metal contamination in soil and water worldwide comes from human activities in developing agriculture and industry such as excessive agrochemicals application, sludge dumping and melting operations, and large-scale effluent water generation, which can be particularly concerning because of the long-term persistence of such toxic metals in the environment [4,5]. 

Heavy metals, on the other hand, are largely unavailable for natural plant uptake; accumulated heavy metals in soils can indirectly cause oxidative damage to plant cells as abiotic stress agents by changing soil environmental conditions (e.g., soil pH, organic matter decomposition, microbial activity, competitive ions adsorption, and so on) [2,6]. Besides that, one major theoretical issue that has dominated the field in recent years concerns climate change and its consequences (e.g., reduced rainfall, rapid and intense weather variability, and high soil temperatures) which can drastically and swiftly alter soil chemical and physical properties and consequently heavy metals mobility and availability [2].

The threat signals emitted by the stress of toxic metals are recognized by the plant and lead to the activation of diverse defense reactions, one of the most important of which is the change in the production of metabolites in the plant tissues as a fundamental detoxification mechanism [6]. Modulating metabolic pathways such as carbohydrates, amino acids, and organic acids metabolisms, can result in relieving heavy metal toxicity by scavenging reactive oxygen species (ROS), precipitating and chelating metals [7]. Therefore, by employing metabolomics qualitative modes of enquiry, it would be possible to illuminate the subtle metabolic alterations in plant tissues under adverse stress and can be used to reflect the abnormalities of all metabolic pathways [8].

Earlier, the authors started a comprehensive research work that provides in-depth analysis to advance the awareness of the harmful effects of sewage sludge, as one of the main sources of heavy metal soil contaminations, on ecosystems. As a part of this project, attempts were made to introduce stress-resistant plant species, orientate their mechanisms of action in exposure to pollution, and evaluate their potential in phytoremediation [5]. Accordingly, *Sesuvium portulacastrum* L. was introduced as one of the stress-tolerant plants with the capability of heavy metals bioaccumulation and bioremediation. Such plants, living in metal-enriched sites, have already been shown to repeatedly increase their tolerance to the metals in question [9]. Apart from the previous research and drawing upon the research into metabolites and biochemical analyses, the main objective of the present research was to reveal the resistance mechanisms activated in *S. portulacastrum* plants under heavy metal toxicity. Until recently, there has been no reliable evidence verifying the functions of metabolites in alleviating heavy metals stress, and much uncertainty still exists. Accordingly, it was hypothesized that carbon and nitrogen metabolisms in the roots and shoots of the *S. portulacastrum* plants change in response to the toxicity of heavy metals. 

## 2. Results

As reported earlier [5], 42 plant species were identified in the polluted areas, among which the most abundant was *Sesuvium portulacastrum*. Although those results revealed the ability of this plant in bioremediation and tolerate the stress of heavy metals in the soil, in order to better understand the mechanisms of action activated by this plant, we measured carbon metabolisms in the roots and shoots of the plants. 

To realize how heavy metals in polluted soils affected the nitrogen metabolism in *S. portulacastrum* plants, we analyzed their amino acid composition. In this regard, glycine, alanine, and lysine were the most abundant amino acids in all sampling sites, which varied from 48.4, 7.9, and 2.3 to 81.5, 14.5, and 6.1 mg g^−1^ of protein in shoot samples, and from 56.8, 9.3, and 3.4 to 66.2, 22.8, and 4.51 mg g^−1^ of protein in roots, respectively (Figure 1). Although the composition of amino acids in all samples was almost the same, their concentrations were obviously different. In confirmation of these findings, Ward’s clustering analysis showed that soil samples from sites with less contamination were clustered in an independent group of contaminated soils in terms of the amino acid compositions in both shoot and root samples (Figure 1). The concentration of some amino acids (e.g., glycine, alanine, glutamate, arginine, ornithine, valine, serine, phenylalanine, and aspartate) in the roots and shoots of *S. portulacastrum* plants grown in contaminated soils were significantly higher than those in the control site, while the concentration of some others (e.g., cysteine) was lower in contaminated soils (Figure 1).

To investigate carbon metabolism, we attempted to illuminate any changes in sugars, organic acids, fatty acids, and tocopherols compositions in *S. portulacastrum* plants in contaminated areas by employing qualitative modes of inquiry of carbon and sugar metabolism. Site 1 reported significantly less starch content in shoot and root samples than the other sites (Figure 1), confirming the negative effect of heavy metals contamination on starch accumulation in plants (Figure 2). 

In the line with these results, the closer we were to the center of contamination, the activity of amylase enzyme, an enzyme that catalyzes the hydrolysis of starch into sugars, increased significantly in the roots and shoots. This increase in site 1 was 2.8 times in shoots and 2.1 times in root compared with the control site (Figure 3). 

On the other hand, total soluble sugar content did not have a clear trend in response to pollution levels (Figure 4). Based on this, the lowest content of soluble sugar in the shoot of *S. portulacastrum* plants was obtained from site 1 (−12% compared with the control site), while its highest concentration in the root belonged to the samples collected from sites 1 and 2 (+48 and +50% compared with the control site, respectively) (Figure 4). Therefore, we tended to focus on the concentration of soluble sugars (glucose, sucrose, and fructose) separately rather than the total amount of soluble sugars. In this regard, the content of fructose and sucrose in shoot tissues in all sampling sites were placed in the same statistical group and did not differ significantly from each other, but the accumulation of glucose in the control site was 27% more than its amount in site 1 (Figure 3). Contrary to shoots, we observed a significant increase in the amount of fructose and sucrose in roots in the contaminated sites, so that their amount in site 1 was 75% and 204% higher than the control site, respectively. Unlike glucose in shoots, this study did not find a significant difference in glucose accumulation in roots between the control site and site 1. Moreover, in the same proportion as increased activity of sucrose phosphate synthase, an enzyme involved in sucrose biosynthesis in the shoots and roots of *S. portulacastrum* plants in contaminated areas increased, the increase in sucrose hydrolyzing enzymes (vacuolar invertase and cytosolic invertase) also increased significantly in contaminated sites compared with the control site (Figure 3). 

The fatty acid composition was evaluated to show the possible changes in *S. portulacastrum* plants grown in the contaminated soils compared with the control site (Figure 5). Sixteen fatty acids were detected in both shoot and root samples, the most concentrated of which was oleic acid (octadecenoic; C 18:1) as an unsaturated fatty acid, equal to 62.4 ng g^−1^ in shoots and 60.5 ng g^−1^ in roots at site 1 (about +29% and +49% compared with those in the control site). Following that, three saturated fatty acids, including palmitic acid (hexadecanoic acid; C 16:0), stearic acid (octadecanoic acid; C 18:0), and arachidic acid (eicosanoic acid; C 20:0), had the highest accumulation in root and shoot samples, which were the concentrations of the palmitic and stearic acids in the contaminated samples were significantly higher than the control site. The accumulation of arachidic acid was non-significantly lower than at the control site (Figure 5).

Although the changes in sigma tocopherol in the roots and shoots of *S. portulacastrum* did not follow a specific trend, it was clear that as we progressed from the control site to the center of contamination, the concentration of alpha and beta tocopherols in the plant, which accounted for more than 99% of the total concentration of tocopherols, increased significantly (Figure 6). Based on this, the concentration of total tocopherols changes in experimental sites 1, 2, 3, and 4 were about +121, +105, +59, and +39% in the shoots, and +66, +53, +28, and +20% in the roots compared with its concentration in the control site (Figure 6). 

The present research also sought to determine any alteration in organic acid composition in *S. portulacastrum* tissues, because of its key role in many biochemical and cellular reactions. Accordingly, six organic acids in all samples were found including malate, succinate, citrate, isobutyric, oxalate, and fumarate, respectively. Contaminated soils significantly increased malate concentrations, while it did not show any specific effect on other organic acids (Figure 7).

## 3. Discussion

The responses of *S. portulacastrum* plants to heavy metals-contaminated soils were investigated by screening nitrogen and carbon-based metabolites. It is apparent from the results that the concentration of almost all detected amino acids in shoot and root samples were affected by soil heavy metals contamination. Accordingly, a further increase in the content of some amino acids (such as glycine, alanine, glutamate, arginine, ornithine, valine, serine, phenylalanine, and aspartate) suggests the possibility that they are specifically secreted to adapt to external heavy metal stress. This finding supports previous research into this area which links changing amino acid content and inducing a high level of heavy metals tolerance [1,10]. It is a widely held view that such amino acids, especially proline, glycine, and alanine, are involved in stabilizing macromolecules structures, scavenging free radicals, adjusting heavy metal toxicity by binding metal ions inside cells of several plant tissues, and altering the solubility, adsorption and desorption, fractions, and migration of metals through dissolution, chelation, and oxidation/reduction [11,12,13].

In the present study, apart from malate, the rest of the detected organic acids did not show a specific reaction to the heavy metals contamination. These results somehow do not agree with the findings of other studies, in which the levels of accumulated and secreted organic acids in plants were considerably increased in response to heavy metals stress [8,12,14]. Some previous metabolic analyses have revealed that plants growing in heavy metals-contaminated soil can secrete organic acids, especially citrate and oxalate, from their roots to effect on metal mobilization and absorb them [8,15,16]. Since the accumulation of heavy metals in *S. portulacastrum* was observed in our previous research [5], it might be concluded that among the organic acids, only the increased secretion of malate from the root changed the acidity of the contaminated soil and made more of these toxic metals available to the plant. In addition, malate content in the shoot and root of *S. portulacastrum* is therefore likely to contribute to reducing the effect of toxicity experienced by plants as a chelating agent in binding to toxic metal ions in the rhizosphere or within the apoplast, and then preventing them from entering the cytoplasm [17].

Under metal toxicity, some unsaturated fatty acids such as oleic (C 18:1), the most concentrated fatty acid, and palmitoleic (C 16:1) were more accumulated in the shoot and secreted from the root of *S. portulacastrum* than in non-contaminated plants. This finding seems to be consistent with other research, which found excitation of the synthesis of the unsaturated fatty acids and (or) their release from membrane structures as a plant defense mechanism against abiotic stresses such as heavy metals [18], which can fortify cell wall/membranes and defend cell from lipid peroxidation [19]. Furthermore, increasing the concentration of some saturated fatty acids such as palmitic (C 16:0) and stearic (C 18:0) in contaminated sites was somewhat unanticipated. A possible explanation for this might be that the release of some metabolites from the roots (e.g., malate and amino acids) by changing the conditions in the rhizosphere increased the absorption of nutrients such as iron and potassium, which are required for the synthesis of saturated fatty acids and the activation of enzymes involved in it [20].

The increasing trend in alpha and beta tocopherols content in the shoot and root of *S. portulacastrum* under increased heavy metal contamination in soil corroborates the earlier findings reported by Mishra et al. [21] and De Agostini et al. [22]. Under heavy metal toxicity, the byproducts of these vitamins E, as non-enzymatic antioxidant compounds, are significant in at least major respects of scavenging ROS and preventing lipid peroxidation within plant cells, thereby protecting plant cellular damage [23,24]. Moreover, the findings of the current study on the alterations of carbohydrate metabolisms in *S. portulacastrum* plants under heavy metals toxicity are consistent with those of Hejna et al. [25] and Wang et al. [26] who reported changing in carbohydrate accumulation and distribution in plants in the presence of heavy metals in soil and proved their multifunctionality in ROS detoxification and stress signal transmission. Increased amylase accumulation in the shoot and root of *S. portulacastrum* plants grown in contaminated soils resulted in a decrease in starch content, particularly in the shoots, because this enzyme catalyzes the hydrolysis of glycosidic bonds in starch, breaking it down and converting it to simple sugars [26,27]. Despite the significant increment in the accumulation of sucrose in the roots of plants grown in the contaminated sites compared with the shoots, the activity of sucrose hydrolyzing enzymes, including vacuolar invertase and cytosolic invertase, in both shoots and roots remained almost equal. The reason for this is not clear but it might have something to do with the translocation of sucrose from shoot to root under stress. This eliminated the need for more hydrolyzing sucrose in the shoots, and then led to no increase in the glucose and fructose content in the shoots. Moreover, the observed increase in the accumulation of sucrose phosphate synthase in *S. portulacastrum* plants can be attributed to the activation of the pathway of converting glucose and fructose into sucrose, which is not broadly consistent with the earlier idea that the activities of the sucrose enzymes involved the biosynthesis direction were decreased under stress [28]. As a result, it is possible to speculate that the other active defense mechanisms described in this study played a larger role in protecting plant cells from ROS damage caused by heavy metal stress, and that carbohydrates metabolism passively preserved the fundamental conditions for physiological activities in heavy metals-resistant plants [28]. 

## 4. Materials and Methods

### 4.1. Experimental Set up

A total of 75 *S. portulacastrum* plants at the same growing stage were taken from 5 different experimental sites (5 subsamples × 5 sites × 3 replicates) around man-fabricated sewage dumping lake at Jeddah (Saudi Arabia, located at 21°35′ N, 39°19′ E, at an elevation of 150 m above sea level; and a total of 40 mL, and an average of 29.3 °C, and 60.6% of long-term annual precipitation, temperature, and relative humidity, respectively), where around 5 × 104 m^3^ day^−1^ of effluent is transported to the lake [5]. *S. portulacastrum* plant, as an herbaceous and perennial, is one of many alien plant species abundant in the western region of Saudi Arabia [29]. This plant can grow naturally in the sub-tropical and Mediterranean areas, not only because of the reliance on its potential root system but also due to the molecular and physiological flexibility to adapt to various abiotic stress [29,30]. 

Five experimental areas, each with 25,000 m^2^, were chosen for sampling where soils were not disturbed by land usage, livestock grazing, roads, or other pollution sources. The first, second, third, and fourth sites (S1, S2, S3, and S4) were 50, 100, 500, and 1000 m away from the sewage dumping lake, respectively, while the non-polluted site (control site) is located 5000 m away from the lake. Soils in all sites were sandy loam with pH 7.3–7.8. The soil electrical conductivity (EC) and organic matter ranged from 0.71 ds m^−1^ and 0.72% in the control site to 1.05 ds m^−1^ and 1.94% in S1 (the most polluted area). The concentration of 11 detected heavy metals is presented in Table 1 [5]. To measure the negative effect of heavy metals on microbial population in soil, ten grams of each soil sample was added to 95 mL of 0.1% (*w/v*) solution of sodium pyrophosphate. After homogenization for 30 min, this solution was decimally diluted (10^−1^ to 10^−7^) and aliquots of the resulting solutions plated on appropriate culture media. After incubation at 25 or 30 °C, for up to 10 days, the colony-forming units (CFUs) were counted.

### 4.2. Measurement of Amino Acid Profile

For the extraction of amino acids, 100 mg of shoot and root samples were homogenized in 80% aqueous ethanol for 1 min at 7000 g, spiked with norvaline, and centrifuged at 14,000× *g* for 20 min. The clear supernatant was vacuum-evaporated, and then the particle was resuspended in chloroform. The residual was then again extracted using HPLC-grade deionized water, centrifuged one more time, and the supernatant was mixed with the pellet suspended in chloroform. The centrifuged aqueous phase was filtered using a Millipore microfilter with a pore size of 0.2 M. (14,000× *g*, 10 min). We quantified amino acids using a Waters Acquity UPLC-tqd system (Milford) and a BEH amide column [31].

### 4.3. Carbohydrate Extraction and Estimation

The separation of soluble sugars was carried out in ethanol (80% vol/vol) at 80 °C for 60 min, followed by the addition of freshly prepared anthrone reagent (150 mg anthrone in 100 mL H_2_SO_4_ [72%]), which was then heated in a water bath at 100 °C for 10 min before cooling in an ice bath for 5 min. After soluble sugar extraction, the residual pellet’s starch concentration was measured [32]. The starch solution was hydrated, 90% gelatinized, precipitated, washed with ethanol, centrifuged, and vacuum-dried at 30 °C before being extracted with amylase and amyloglucosidase. By measuring their absorbance at 625 nm, a multi-mode microplate reader (Synergy Mx, Biotek, Santa Clara, CA, USA) was used to calculate total soluble and insoluble sugar [33].

### 4.4. Organic Acid Analysis

A known weight of ground shoot and root samples (about 100 mg) was used, according to AbdElgawad et al. [34], for the quantitative evaluation of specific organic acids. Butylated hydroxyanisole was added to phosphoric acid (0.1%) used to extract organic acids, which were subsequently centrifuged at 14,000× *g* for 30 min at 4 °C. The supernatants were put through Millipore microfilters with a pore size of 0.2 M before being put through an isocratic HPLC column with 0.001 N sulphuric acid at a flow rate of 0.6 mL per minute. The experiment was conducted using the Ultimate 3000 RSLC nano HPLC system. Similar to that, the separation was performed at 65 °C with an Aminex HPX-87 H (310 mm 7.7 mm) column and a Bio-Red IG Cation H (30 4.6) pre-column.

### 4.5. Assessment of Fatty Acid Profile

Following the procedure outlined by Torras-Claveria et al. [35], fatty acids were extracted and quantified. In a nutshell, methanol was applied to 100 mg of grain samples at room temperature until the samples were discolored. Codeine and nonadecanoic acids were then added as internal standards. On a Hewlett Packard 6890, MSD 5975, a gas chromatography-mass spectrometry (GC-MS) analysis was performed. The NIST 05 database and the Golm Metabolome Database (http://gmd.mpimp-golm.mpg.de (accessed on 10 February 2022) were used to identify fatty acids.

### 4.6. Determination of Tocopherol Content

According to the procedures outlined by AbdElgawad et al. [36], tocopherols were extracted in n-hexane solvent and measured by HPLC (Shimadzu, Kyoto, Japan) under normal phase conditions (Partisil Pac 5 m column material, length 250 mm, i.d. 4.6 mm). As an internal standard, dimethyl tocol (DMT; 5 ppm) was also applied. The HPLC system’s included Shimadzu Class VP 6.14 software was used to analyze the data.

### 4.7. Statistical Analysis

The graphs were created using SigmaPlot (SigmaPlott v11.0, Systat Software Inc., San Jose, CA, USA), which was also used to conduct statistical analyses including a two-way analysis of variance and the Tukey’s HSD test (honestly significant difference). Ward’s clustering study was carried out using the NCSS tool (Version 21.0.3.).

## 5. Conclusions

The empirical findings in this study provide a new understanding of defense mechanisms employed by *S. portulacastrum* as a metals-resistant plant by metabolomics analysis. The significant findings to emerge from this research were the modifications of some nitrogen and carbon metabolisms in plant tissues, which can be one of the main mechanisms of the plant in resisting heavy metals toxicity. In detail, the antioxidant defense system emerged as reliable protection for plants in contaminated soils, the most important of which included the accumulation of tocopherols (alpha and beta tocopherols) in the plant tissues. In addition, the alterations in some metabolites accumulation and secretion in plants allowed greater quantities of some low molecular weight compounds such as amino acids (glycine, alanine, glutamate, arginine, and ornithine), organic acid (malate), unsaturated fatty acids (oleic and palmitoleic), saturated fatty acids (palmitic and stearic), simple sugars (sucrose and fructose in root), and the enzymes activities involved in their metabolisms to be utilized to resist metal stress. In general, therefore, it seems that these mechanisms may be cooperative and complementary to induce resistance in the plants in coping with stress in contaminated soils.

Further research is needed to estimate the amount of secretion of these metabolites from the root to the rhizosphere using the same experimental setup, to examine more closely the links between the accumulation of metabolites in the root and their secretion from the root.

## Figures and Tables

**Figure 1 plants-12-00676-f001:**
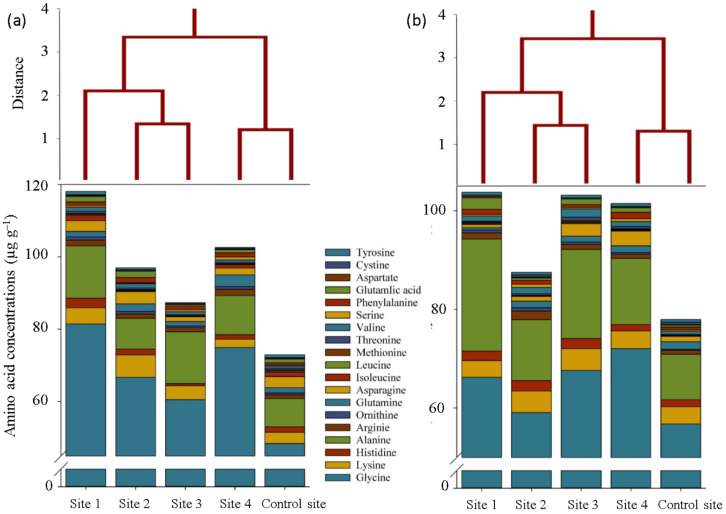
Amino acid composition and related cluster analyses (Ward’s method) in the shoot (**a**) and root (**b**) of *S. portulacastrum* plants at different levels of heavy metals pollution.

**Figure 2 plants-12-00676-f002:**
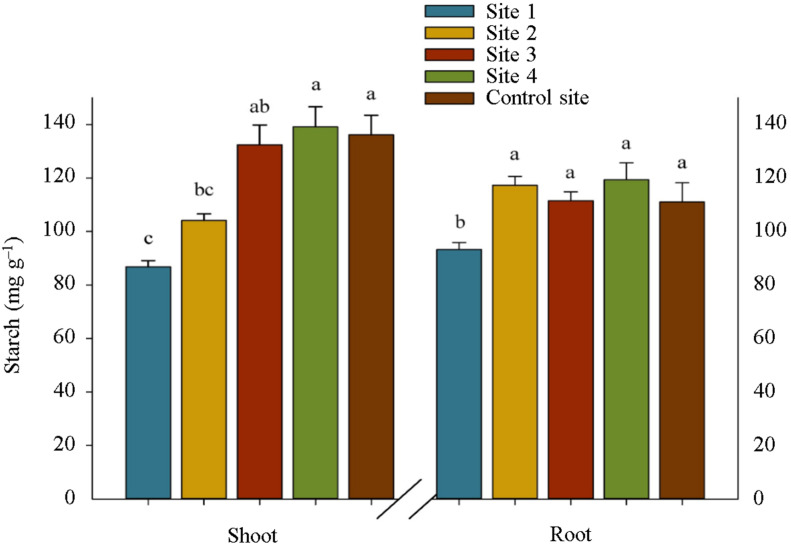
Starch content in the shoot and root of *S. portulacastrum* plants at different levels of heavy metals pollution. Means in each tissue followed by similar letter(s) are not significantly different at 5% probability level (Tukey’s HSD test).

**Figure 3 plants-12-00676-f003:**
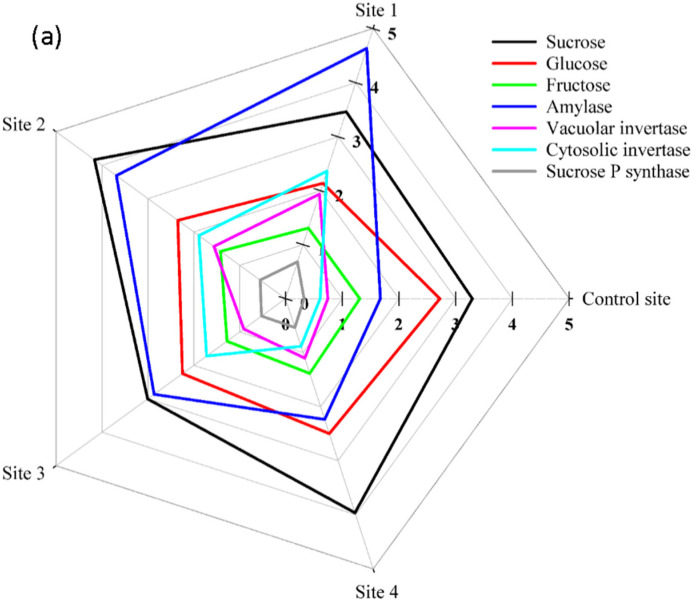
The concentration of sucrose, glucose, fructose (mg g^−1^), and the enzymes involved in their metabolism and biosynthesis (pmol ml^−1^) in the shoot (**a**) and root (**b**) of *S. portulacastrum* plants at different levels of heavy metals pollution.

**Figure 4 plants-12-00676-f004:**
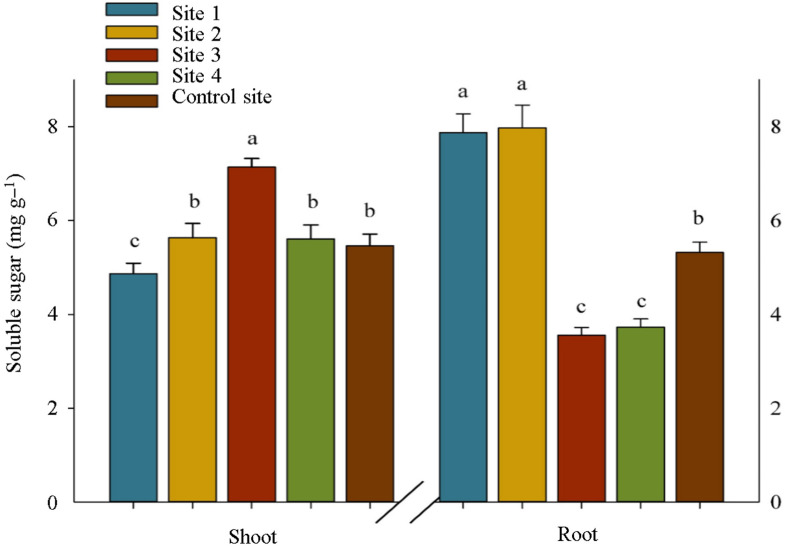
Soluble sugar content in the shoot and root of *S. portulacastrum* plants at different levels of heavy metals pollution. Means in each tissue followed by similar letter(s) are not significantly different at 5% probability level (Tukey’s HSD test).

**Figure 5 plants-12-00676-f005:**
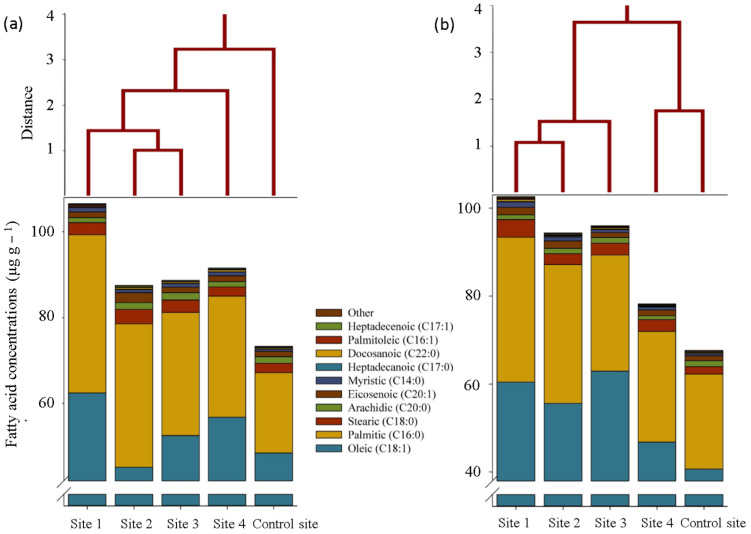
Fatty acid composition and related cluster analyses (Ward’s method) in the shoot (**a**) and root (**b**) of *S. portulacastrum* plants at different levels of heavy metals pollution.

**Figure 6 plants-12-00676-f006:**
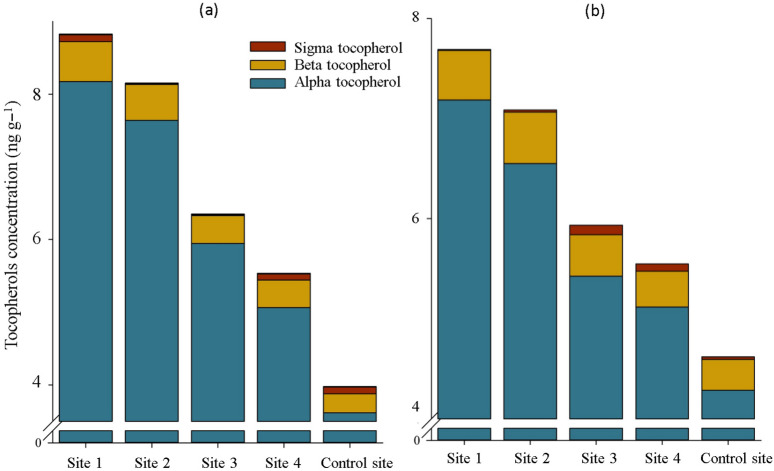
Tocopherols content in the shoot (**a**) and root (**b**) of *S. portulacastrum* plants at different levels of heavy metals pollution. Means in each tissue followed by similar letter(s) are not significantly different at 5% probability level (Tukey’s HSD test).

**Figure 7 plants-12-00676-f007:**
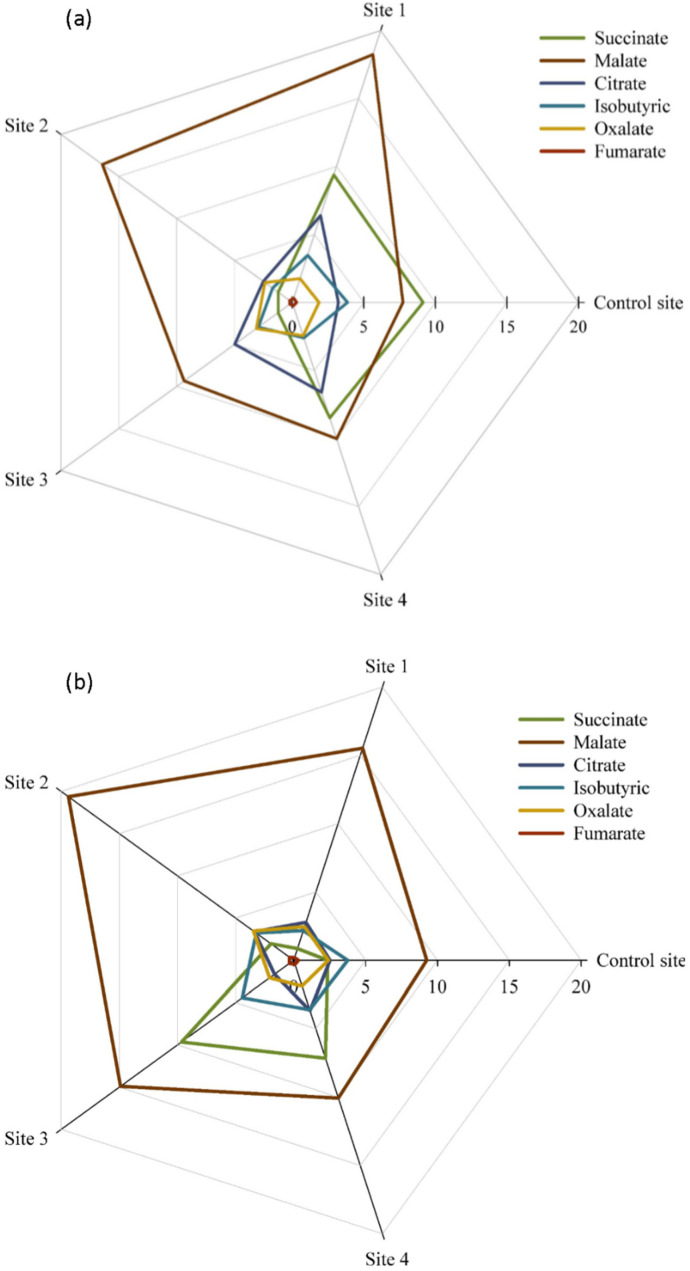
The concentration of organic acids (ng g^−1^) in the shoot (**a**) and root (**b**) of *S. portulacastrum* plants at different levels of heavy metals pollution.

**Table 1 plants-12-00676-t001:** Concentration of heavy metals in soils at control experimental sites, and their changes (%) in the polluted sites as compared with the control site.

	Control Site	S4	S3	S2	S1
Cd	0.43 (µg g^−1^)	+239%	+608%	+1054%	+1107%
Ni	0.441 (µg g^−1^)	−9%	+82%	+153%	+186%
As	0.461 (µg g^−1^)	−27%	−50%	−24%	−2%
Cu	0.863 (µg g^−1^)	+56%	+29%	+56%	+37%
Pb	0.072 (µg g^−1^)	+1503%	+6475%	+10,189%	+14,761%
Co	0.108 (µg g^−1^)	+595%	+3527%	+5240%	+7928%
Hg	0.088 (µg g^−1^)	+309%	+3036%	+6483%	+6320%
Al	0.046 (µg g^−1^)	+761%	+602%	+1267%	+1465%
V	0.046 (µg g^−1^)	+585%	+922%	+1383%	+1683%
Cr	0.06 (µg g^−1^)	+335%	+928%	+1125%	+1600%
Zn	0.046 (µg g^−1^)	+109%	+1191%	+1587%	+1835%
Total bacteria	232	−11%	−18%	−21%	−37%
Spore forming bacteria	47.2	−5%	−8%	−15%	−27%
Actinobacteria	33.5	−17%	−23%	-37%	−35%
Total Fungi	45.8	−21%	−28%	−36%	−45%

## Data Availability

The data presented in this study are available upon request from the corresponding author.

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
