# Peer review of "Understanding the Active Mechanisms of Plant (*Sesuvium portulacastrum* L.) against Heavy Metal Toxicity"

_plants, 2023, doi:10.3390/plants12030676_

Round 1
Reviewer 1 Report
The present manuscript is an outcome of extensive study but before acceptance it requires further changes to broader acceptability.
1. Abstract portion is not clear. It should be modified by presenting clear objective, brief experimental design and major outcome of the study.
2. Add few more key words.
3. In introduction part clearly indicate the novelty of the work.
4. Graphical representations are very nice but legends should be more clear.
5. Add few more recent references all through the manuscript.
6. Check typos and grammar.
Author Response
The authors would like to thank the reviewer for their precious time and valuable comments. We have carefully addressed all the comments. The corresponding changes and refinements made in the revised paper are summarized in our response below.
Reviewer's comments to the author:
Referee: 1#
Abstract portion is not clear. It should be modified by presenting clear objective, brief experimental design and major outcome of the study.
Response: We have modified the abstract based on the reviewer’s comment.
- Add few more key words.
Response: We have changed the keywords as follows:
“Keywords: Amino acid; Antioxidant; Fatty acid; Heavy metal; Metabolite; Sesuvium portulacastrum L.; Tocopherol.”
- In introduction part clearly indicate the novelty of the work
Response: In order to accommodate this comment, we rewrote the objective and hypothesis section as follows:
“Apart from the previous research and drawing upon the stand of research into metabolomics analysis, the main objective of the present research was to reveal the resistance mechanisms activated in S. portulacastrum plants under heavy metal toxicity. Until recently, there has been no reliable evidence verifying the functions of metabolites in alleviating heavy metals stress, and even much uncertainty still exists. Accordingly, it was hypothesized that carbon and nitrogen metabolisms in the roots and shoots of the S. portulacastrum plants change in response to the toxicity of heavy metals.”
- Graphical representations are very nice but legends should be more clear.
Response: Thanks for the comment. We re-arranged the legends and improved the quality of the figures to be clearer.
- Add few more recent references all through the manuscript.
Response: 22 references out of 34 listed references have been published in the last five years. Many of the older references are related to methods introduced in metabolite analysis by researchers in distant years, which are mentioned in the M&M section. Nevertheless, two new references have been added to the M&M section as follows:
Alharthi,S.T.; El-Sheikh, M.A.; Alfarhan, A.A. Biological change of western Saudi Arabia: Alien plants diversity and their relationship with edaphic variables. Journal of King Saud University - Science 2023, 35, 102496. https://doi.org/10.1016/j.jksus.2022.102496
Ding, G.; Yang, Q.; Ruan, X.; Si, T.; Yuan, B.; Zheng, W.; Xie, Souleymane, O.A.; Wang, X. Proteomics analysis of the effects for different salt ions in leaves of true halophyte Sesuvium portulacastrum. Plant Physiology and Biochemistry 2022, 170, 234-248. https://doi.org/10.1016/j.plaphy.2021.12.009
- Check typos and grammar.
Response: As suggested by the reviewer, we have made all necessary changes in the text and we tried simultaneously, to improve the written English language.
Reviewer 2 Report
The paper does not contain new data or new ideas. The scientific problem is not well stated; e.g., there is no hypothesis. The reader should know from the beginning what metal ions the plants were treated with, and above all, what the research's purpose was. At the end of the article, we learn that we are dealing with soil taken from a polluted environment. The authors use the terms metal and metal interchangeably. Although the methods are suitable and adequately described, there are no conditions for plant cultivation. In my opinion, the authors did not present that the experiments are properly planned and executed. The title and abstract of the manuscript are not pertinent and understandable. The manuscript should be deeply corrected.
Author Response
The authors would like to thank the reviewer for their precious time and valuable comments. We have carefully addressed all the comments. The corresponding changes and refinements made in the revised paper are summarized in our response below.
Referee: 2#
- The paper does not contain new data or new ideas. The scientific problem is not well stated; e.g., there is no hypothesis.
Response: Thanks for the comment. As we mentioned above, we have re-written the objective and hypothesis to describe our research clearer.
- The reader should know from the beginning what metal ions the plants were treated with, and above all, what the research's purpose was.
Response: We slightly disagree with the reviewer comment. In the M&M section, it was fully explained about the contamination of the experimental sites with 10 heavy metals (including Cd, Ni, Cu, Pb, Co, Hg, Al, V, Cr, Zn) out of 11 studied metals, as shown in Table 1. So the plants were exposed to a combination of heavy metals and not just one.
- At the end of the article, we learn that we are dealing with soil taken from a polluted environment.
Response: Sorry, but we didn't understand what the reviewer meant. In the entire text of the manuscript, we repeatedly mentioned plants were grown in contaminated areas (soils) with heavy metals. Therefore, this was not something that was only mentioned at the end of the manuscript.
- The authors use the terms metal and metal interchangeably.
Response: We are sorry, the comment is not clear to us to be responded.
- Although the methods are suitable and adequately described, there are no conditions for plant cultivation.
Response: Thanks for the accurate comment. S. portulacastrum plant has not been cultivated in contaminated areas. Indeed this plant, as an herbaceous, perennial, and facultative halophyte, grows naturally in the sub-tropical, Mediterranean, and warmer areas around the world, relying on its potential root system. As mentioned in the M&M section, plants were collected at a similar growth stage (flowering).
Lokhande V, Gor B, Desai N, Nikam T, Suprasanna P. 2013. Sesuvium portulacastrum, a plant for drought, salt stress, sand fixation, food and phytoremediation. A review. Agronomy for Sustainable Development, 33 (2), pp.329-348. 10.1007/s13593-012-0113-x
According to the reviewer comment, we have added the following paragraph to the M&M section:
“S. portulacastrum plant, as an herbaceous and perennial, is one of many alien plant species abundant in the western region of Saudi Arabia [29]. This plant can grow naturally in the sub-tropical and Mediterranean areas, not only because of the reliance on its potential root system but also due to the molecular and physiological flexibility to adapt to various abiotic stress [29, 30].“
- In my opinion, the authors did not present that the experiments are properly planned and executed.
Response: Sorry again, this comment is not clear as well and contradicts the reviewer's previous comment a bit. We re-checked the M&M section, but we did not understand what we should have mentioned regarding the plan and execution of the experiment.
- The title and abstract of the manuscript are not pertinent and understandable.
Response: We have revised the title as follows:
“Understanding the active mechanisms of Plant (Sesuvium portulacastrum L.) against Heavy Metal Toxicity”
We have also revised the abstract based on the comments.
- The manuscript should be deeply corrected.
Response: It would be great if we could make the corrections desired by the reviewer, which cannot be addressed here, as this comment is very general. However, the sections mentioned by both reviewers such as title, abstract, objectives, and M&M were modified.
Round 2
Reviewer 2 Report
Article has been corrected according to my suggestion; so I’d like to recommend accepting to publication in Plants.